# OpenReview forum: "Jailbreak Antidote: Runtime Safety-Utility Balance via Sparse Representation Adjustment in Large Language Models"
_ICLR.cc/2025/Conference — ICLR 2025 Poster_

### Official Review · Reviewer_FZR8 · 2024-10-27

**Soundness:** 2
**Presentation:** 3
**Contribution:** 2
**Rating:** 6
**Confidence:** 4

**Summary:**

This paper proposes a defense method against LLM jailbreaking attacks, which operates at model inference-time by manipulating a sparse subset of the model's representations (internal states). Specifically, the internal states are moved towards pre-computed safe directions, which make the model's responses safer against harmful requests. This method enables better model safety without leading to significant utility drop.

**Strengths:**

* This proposed methodology to defend against jailbreaking attacks is neat. The paper is well structured and written.
* The finding of the sparsity (5\%) of safety representations is intriguing.
* The results are promising -- largely diminishing the success rate of jailbreaking attack while maintaining high utility, which surpass the compared baseline defenses.

**Weaknesses:**

* My major concern is the incremental contribution of this work to [1] and [2]. For example, in [1], the authors already provided extensive and thorough study regarding representation engineering, and discussed methods to adjust model behaviors against LLM jailbreak attacks. These existing work largely diminish the contribution of this paper.

* In line 78-79, you argued that safety alignment methods "lacks real-time flexibility." But on the other hand, you also did not highlight the benefit of "real-time flexibility" on your method. Then this point should not be emphasized too much.

* Should include more defense baseline comparison. I appreciate that the authors considered 6 existing jailbreak defense baselines which operate mostly at inference time, but I think it would be beneficial to consider safety alignment defenses as well, e.g., [3] and [4].

  In Line 79, you mentioned that these methods "*may* reduce model utility," but you didn't compare them with your method throughout the paper. Do they?

* In "Inference Efficiency Analysis" (page 8), the authors used "the number of defense tokens required" as a metric to compare method efficiency. This is clearly a bad metric for efficiency study -- compared to additional output tokens, additional input tokens induce less significant overhead. Thus, it's not clear whether the proposed method is indeed more efficient than the baseline methods. I suggest the authors use actual GPU time instead of number of defense tokens as the metric here.

  Also, could you elaborate on the (pre-inference) time cost to obtain the safety representations and masks?

* There should be some adaptive analysis against the proposed defense. That is, you should consider if an adversary is already aware that the victim model is deployed with your proposed defense. Then, if the adversary explicitly optimize for an adversarial prompt $S'$ against your safety representations, whehter / how much can this adaptive attack be defended by Jailbreak Antidote? This is an necessary study that should come alongside the proposed defense.

[1] Representation Engineering: A Top-Down Approach to AI Transparency, Arxiv 2023

[2] Steering Language Models With Activation Engineering, Arxiv 2024

[3] Improving Alignment and Robustness with Short Circuiting, NeurIPS 2024

[4] Safety Alignment Should Be Made More Than Just a Few Tokens Deep, Arxiv 2024

**Questions:**

* I think there need to be more specifications and analysis on Fig 2(a) and Sec 4.2. For example, how is this figure plotted? And how did you come to the conclusion that "the safety representation in LLMs is sparse"?

  From my understanding, most entries in each $\textbf{d}^l_{safe}$ are almost 0, which indicates *sparsity*. Then, what's the principle behind masking these safety representations (Eq 4)? Since most entries are already 0, in Eq 5, it doesn't make a difference whether you mask them or not, right?

* Are the DSP values reported in Table 1 averaged over all jailbreaking methods?

* Why the method works better for larger models? I see there are some discussions on this. But could you also perform some more in-depth studies (e.g., analyzing the representation distribution difference between small and large models) to provide more understanding about this?

* Could you also discuss the relationship of your observation of sparsity in safety representations, with [1]'s observation of sparsity in model weights that account for safety?

[1] Assessing the brittleness of safety alignment via pruning and low-rank modifications, ICML 2024

---

> ### Author Response · Authors · 2024-11-22
> **Rebuttal - Part I/III**
>
> *We appreciate the reviewer's recognition of the novelty of our method and the promising results. We address your concerns below.*
>
> 1. **Incremental Contribution Relative to Prior Work**
>
>    *Response*: Thank you for raising this point and for referencing these excellent works. While our paper shares some conceptual similarities with [1] and [2] in exploring the internal states of LLMs, there are significant differences that we would like to highlight:
>
>    1. **Distinct Focus**: Unlike [1] and [2], which primarily investigate LLM internal states from a representation perspective, our primary objective is to develop a low-overhead, practical method to enhance LLM safety, specifically targeting defenses against jailbreak attacks. Our extensive experiments validate the effectiveness of our proposed Jailbreak Antidote in achieving this goal. This focus on practical safety applications sets our work apart.
>
>    2. **Novel Contributions to Representation Methodology**: As you pointed out, we uncover the sparsity of safety-related representations in LLMs—a novel and significant finding. This sparsity allows us to adjust only a small subset of neurons (~5%) to achieve substantial safety improvements. While [1] and [2] provide foundational insights into LLM representations, they do not address or explore this sparsity aspect, particularly in the context of safety. We believe this finding contributes meaningfully to the understanding of LLM representations and could inspire further advancements in model interpretability.
>
>    3. **Comprehensive Experiments and Demonstrated Effectiveness**: Our work includes extensive experiments across a wide range of tasks and settings, showcasing the robustness and utility of our approach. The demonstrated success of Jailbreak Antidote in balancing safety and utility further underscores the practical relevance and impact of our contributions.
>
>     We have updated the related work section to emphasize these differences and better clarify how our contributions build upon and diverge from prior work. Thank you for providing valuable feedback that helped us improve the framing of our contributions.
>
>
> 2. **Emphasis on Real-Time Flexibility**
>
>    *Comment*: "You argued that safety alignment methods 'lack real-time flexibility'... But you also did not highlight the benefit of 'real-time flexibility' on your method."
>
>    *Response*: We apologize for not making this clear. Our method allows for real-time adjustment of the safety-utility trade-off through the scaling factor $\alpha$, enabling dynamic control based on context or user preferences (Section 3.4 and Figure 5). Unlike methods that require retraining or fine-tuning, our approach enables users to adjust safety levels on-the-fly without reloading the model, providing significant flexibility.
>
> 3. **Comparison with Safety Alignment Defenses**
>
>    *Comment*: "Should include more defense baseline comparison. I appreciate that the authors considered 6 existing jailbreak defense baselines... but I think it would be beneficial to consider safety alignment defenses as well."
>
>    *Response*: We agree and have included comparisons with safety alignment defenses, such as preference-based fine-tuning approaches [7, 9]. The results, presented in Table 2 and also in `Table R.1` in the General Response, demonstrate that our method achieves higher defense success rates while effectively balancing safety and utility. Additionally, we compared against SafeDecoding [3] specifically in the Response to Reviewer `EqZp`, further illustrating the robustness of our approach.
>
> 4. **Efficiency Metrics**
>
>    *Comment*: "In 'Inference Efficiency Analysis'... the authors used 'the number of defense tokens required' as a metric... It's not clear whether the proposed method is indeed more efficient than the baseline methods."
>
>    *Response*: We understand your concern. We have added actual inference time measurements to provide a more concrete comparison of efficiency. The results, shown in Table A.7 of the appendix (`Table R.4` in the General Response), indicate that our method introduces minimal computational overhead during inference, often even reducing inference time due to shorter responses resulting from improved safety.
>
> 5. **Elaboration on Pre-Inference Time**
>
>     *Comment*: "Could you elaborate on the (pre-inference) time cost to obtain the safety representations and masks?"
>
>     *Response*: Certainly. We have included the pre-inference time for computing the safety directions in Table A.6 of the appendix (`Table R.3` in the General Response). The pre-inference computation is efficient, taking only a few minutes even for the largest models.

---

> > ### Author Response · Authors · 2024-11-22
> > **Rebuttal - Part II/III**
> >
> > 6. **Adaptive Attack Analysis**
> >
> >     *Comment*: "There should be some adaptive analysis against the proposed defense..."
> >
> >     *Response*: We have conducted experiments to evaluate our method against adaptive attacks where the adversary is aware of our defense mechanism. We included attacks like GCG [8], PAIR [4], and AutoDAN [6] (`Table R.2` in the General Response). The results in Table A.4 demonstrate that our method remains robust under these adaptive scenarios.
> >
> >
> > 7. **Clarifications on Figures and Sparsity**
> >
> >     *Comment*: "I think there need to be more specifications and analysis on Fig 2(a) and Sec 4.2..."
> >
> >     *Response*:  Figure 2(a) is generated using dimensionality reduction with `sklearn`. Specifically, we used various prompts to query the LLM, recorded its internal states, and then performed dimensionality reduction. Figure 2(a) demonstrates that shallow layers correspond to low-level features like syntax and structure, while deeper layers encode high-level conceptual features. At the deepest layers, prompts from adversarial attacks are observed to transition between *Benign* and *Harmful* clusters. This observation underpins our method: by translating internal states, the model can respond more cautiously and shift towards safer behavior in the face of adversarial attacks.
> >
> >     The sparsity of safety representations is more directly related to Figure 2(b). This figure presents the distribution of the safety direction vector $d^{l}_{safe}$ derived using Eq. (5). We observe that this vector follows a long-tailed distribution, with most values concentrated near zero, suggesting that safety-related neurons may be sparse. Appendix Figure A.2 further visualizes the safety vectors for various models, supporting this hypothesis.
> >
> >     Building on this, we implemented interventions to adjust a small subset of neurons. As shown in Figure 5, modifying only the top 5% of neurons significantly alters the model’s safety preference while preserving utility. Additionally, Figure 6 demonstrates that selecting the top 5% of neurons yields better results than randomly selecting 5%, further validating our conclusion.
> >
> >     To clarify these findings, we have added more detailed descriptions of Figures 2(a) and 2(b) in the manuscript. We hope this explanation addresses your concerns.
> >
> > 8. **Principle Behind Masking Safety Representations**
> >
> >    *Comment*: "Since most entries are already 0... it doesn't make a difference whether you mask them or not, right?"
> >
> >    *Response*: While many components are close to zero, masking the negligible ones is crucial to prevent unnecessary adjustments that could negatively impact the model's utility. By focusing on the top 5% of components with the largest magnitudes, we ensure that we only adjust dimensions that significantly influence safety. Our ablation studies (Figure 6) show that adjusting only the most significant components yields better performance than adjusting all components.
> >
> > 9. **Are DSR Values Averaged Over All Jailbreaking Methods?**
> >
> >    *Comment*: "Are the DSR values reported in Table 1 averaged over all jailbreaking methods?"
> >
> >    *Response*: Yes, the Defense Success Rate (DSR) values reported are averaged over all jailbreak attack methods to provide a comprehensive assessment of the defenses' effectiveness across diverse attacks. We clarified this in the manuscript (Section 5.2).
> >
> > 10. **Why the Method Works Better for Larger Models**
> >
> >     *Comment*: "Why the method works better for larger models? ... Could you also perform some more in-depth studies..."
> >
> >     *Response*: Larger models generally have greater capacity to encode safety-related information and capture complex patterns, making our adjustments more effective. Exploring the relationship between model size, internal representations, and safety adjustment effectiveness is an interesting direction for future work.
> >
> >
> > 11. **Relation to Observations in Prior Work**:
> >
> >     *Response*: *Response*: Thank you for the insightful connection. We have added a discussion in the related work section to clarify this point. While [2] focus on reducing safety alignment by pruning specific neurons, our work primarily explores how to enhance safety. As shown in Figure 5 and Figure A.5, adjusting our parameter \$\alpha$ to values less than zero could also be used to diminish LLM safety. However, this is not the focus of our study, as we argue that controlling the internal processes of an LLM to facilitate attacks is relatively straightforward. Instead, we address the more challenging problem of improving defense mechanisms.
> >
> >     Moreover, our approach provides the advantage of continuous adjustment of internal states via $\alpha$, as opposed to the binary nature of neuron removal or retention. This flexibility enables fine-grained control, which is critical for balancing safety and utility. We hope this addition clarifies the distinction and complements existing work.

---

> > > ### Author Response · Authors · 2024-11-22
> > > **Rebuttal - Part III/III**
> > >
> > > Thank you for your thoughtful feedback, which helped us clarify our contributions, refine comparisons, and improve key analyses. We hope the revisions address your concerns and would greatly appreciate your reconsideration of the manuscript’s rating.

---

> > > > ### Comment · Reviewer_FZR8 · 2024-11-23
> > > > **Thanks for the rebuttal effort**
> > > >
> > > > I appreciate the authors' considerable effort in their rebuttal to address my concerns. However, I still have some follow-up questions to better understand certain aspects of your work:
> > > >
> > > > * Could you elaborate on the experimental setup for the adaptive attacks described in Appendix A.2.7? Specifically, how are these attacks "adaptive" against your defense, and in what ways do they differ from the attacks presented in your main results (e.g., Table A.7)?
> > > >
> > > > * While I recognize the authors’ explanation and justification for comparing efficiency via "defense tokens," I still believe that reporting and comparing actual wall-clock time in the main body would provide a clearer and more practical measure of method efficiency for real-world practitioners. Actual inference time is a more interpretable and actionable metric. If the difference in inference time is minimal or negligible, then the advantage gained from requiring fewer "defense tokens" would also be negligible.

---

> > > > > ### Author Response · Authors · 2024-11-25
> > > > > **Response to Follow-Up Comments**
> > > > >
> > > > > We appreciate your continued interest in our work and thank you for your insightful follow-up questions. We address each of your concerns below.
> > > > >
> > > > > 1. **Elaboration on the Experimental Setup for Adaptive Attacks**
> > > > >
> > > > >    *Response*: Thank you for seeking clarification on this important aspect. In our initial experiments, we primarily focused on **static attacks**, which are transfer attacks where we apply prompts that have successfully jailbroken other models to our models. These prompts are fixed and do not adapt based on the defense mechanisms in place.
> > > > >
> > > > >    The adaptive attacks described in Appendix A.2.7 are **dynamic** in nature. These attacks continuously modify their inputs in an attempt to break our defense. Unlike static attacks, which use fixed prompts regardless of the defense in place, adaptive attacks iteratively refine their strategies to exploit potential weaknesses in the defense.
> > > > >
> > > > >    We initially focused on static attacks due to computational feasibility. Evaluating every combination of attack method, defense method, and test model for adaptive attacks requires extensive computational resources, as it involves numerous inference and evaluation cycles. Our goal was to estimate the general properties of our defense across a broad set of conditions.
> > > > >
> > > > >    However, we recognize the importance of evaluating defenses against adaptive attacks. Therefore, we have included additional experiments involving adaptive attacks in Table A.4 of the revised manuscript. In these experiments, we fixed the parameters in our **Jailbreak Antidote** defense and allowed adaptive methods, including GCG [8], PAIR [4], and AutoDAN [6], to continuously attempt to bypass it. Details about these adaptive attack methods are provided in Appendix A.1.2.
> > > > >
> > > > >    These adaptive attacks differ from the static attacks in that they are designed to specifically target and overcome defense mechanisms like ours. They represent a more challenging and realistic assessment of a defense's robustness.
> > > > >
> > > > >    We have added specific experimental implementation details in Appendix A.1.2 (lines 840-850) to clarify how these adaptive attacks were conducted. This includes descriptions of how we applied the adaptive attacks.
> > > > >
> > > > >    The results, as previously mentioned, demonstrate that our method remains robust even when facing adaptive attacks designed to target our defense mechanism. By including these evaluations, we provide a more comprehensive assessment of the effectiveness of **Jailbreak Antidote** under various adversarial scenarios.
> > > > >
> > > > > 2. **Reporting Actual Wall-Clock Time for Efficiency Comparison**
> > > > >
> > > > >    *Response*: We appreciate your suggestion and agree that reporting actual inference time provides a clearer and more practical measure of efficiency for practitioners. In response, we have updated Section 5.2 of the main manuscript. The **Inference Efficiency Analysis** now presents actual runtime measurements instead of relying solely on the number of defense tokens.
> > > > >
> > > > >    Specifically, we measured the **runtime per query**, representing the average time taken to process a single query during inference, across different defense methods and models. Our results show that *Jailbreak Antidote* achieves the shortest runtime per query across all models, highlighting its efficiency advantage. This is because our method directly adjusts the internal states without introducing additional tokens or modifying the input prompt, thus minimizing computational overhead.
> > > > >
> > > > >    We have replaced Figure 4 with a new figure that illustrates the runtime per query versus the DSR for different defense methods. This provides a visual representation of the trade-off between efficiency and effectiveness for each method. Due to the format constraints, we cannot include the figure here, but we invite you to refer to the revised manuscript for the detailed results.
> > > > >
> > > > >    Additionally, we have moved the original analysis based on the number of defense tokens to Appendix A.2.11. This hardware-agnostic analysis remains useful for comparing methods across different platforms and provides insights into resource consumption that are independent of specific hardware configurations.
> > > > >
> > > > >    We believe this addresses your concern and offers practical insights for real-world applications.
> > > > >
> > > > > ---
> > > > >
> > > > > We hope that these revisions and explanations address your follow-up questions. Your feedback has been invaluable in improving the clarity and practical relevance of our work. We are grateful for your continued engagement and welcome any further questions you may have.

---

> ### Comment · Reviewer_FZR8 · 2024-11-25
>
> Again, I appreciate the authors' clarifications regarding my follow-up comments, as well as the adjustment of Section 5.2 (efficiency analysis). My remaining concern is mostly around the evaluation against (adaptive) attacks.
>
> ---
>
> I still find the setup of the adaptive attacks insufficiently clarified. For instance:
>
> - How was the GCG attack conducted? Did the GCG optimization process account for the adjusted internal states in your defense (e.g., as defined in Eq. 7)?
> - Additionally, what accounts for the differences between the GCG results in Table A.4 and Figure A.7?
>
> In my view, a qualified "adaptive attack" is more like a (white-box) adversary with precise knowledge of the adjustments made to the model states (i.e., parameters such as (${\mathbf d}_\text{safe}^l, \mathbf m^l, \alpha$), explicitly optimizing their attack to counter these defense mechanisms. For example, an adaptive version of the GCG attack could involve explicitly calculating gradients and optimizing against these additional parameters in Eq. 7. However, I do not see any details in your experiments that reflect such "explicit optimization." If I have overlooked this information, could you kindly point it out?
>
> Dynamic attacks without such "explicit optimization" should not be categorized as "adaptive attacks." For example, while PAIR is an existing dynamic black-box attack that optimizes different jailbreak prompts for different models, it is not specifically designed to adapt to and counter the internal state adjustments of your defense. I recommend not to classify such attacks as "adaptive," as their underlying philosophy does not target to evade your specific defense components.
>
> I understand that evaluating against fully adaptive attacks may require significantly more computational resources, but such evaluations are crucial for demonstrating whether the proposed defense can withstand adversaries that specifically target its mechanisms (not just perform well "in general"). It would be acceptable if your defense were shown to fail in these highly adaptive scenarios, but it is important for readers to be made aware of this potential limitation.

---

> > ### Author Response · Authors · 2024-11-26
> > **Response to Follow-Up Comments**
> >
> > We sincerely appreciate your continued engagement and thoughtful comments. Your insights have been invaluable in helping us clarify and improve our work. We address your concerns below.
> >
> > 1. **Clarification on the Setup of Adversarial Attacks**
> >
> >    *Response*: The GCG attack was conducted following the procedure outlined in the original GCG [8], implementing it as a white-box attack. In our experiments, the parameters of our proposed defense, *Jailbreak Antidote*, including the safety directions $\mathbf{d}_{safe}^l$, masks $\mathbf{m}^l$, and scaling factor $\alpha$, are fixed.
> >
> >    The GCG optimization process inherently accounts for the adjusted internal states resulting from our defense, as these adjustments affect the model's forward pass and the gradients computed during the attack optimization. Since the defense modifies the hidden states, the gradients calculated by GCG naturally include the effects of our defense mechanism as defined in Eq. 7. Therefore, the GCG attack operates with full knowledge of our defense adjustments, effectively making it a white-box, adaptive attack against our method.
> >
> >    Regarding the differences between the GCG results in Table A.4 and Figure A.7:
> >
> >    - **Table A.4** presents the results of **direct attacks**, where the adversarial prompts are generated and applied to the same model with the defense mechanism active. This setup evaluates the model's robustness when the attacker specifically targets it using GCG with knowledge of the defense adjustments.
> >
> >    - **Figure A.7**, however, shows the results of **transfer attacks**, where adversarial prompts generated using GCG on one model (e.g., Llama-2-7B-chat) are applied to different models without further adaptation. This assesses the generalizability and transferability of the attacks across models.
> >
> >    The differing results reflect the effectiveness of our defense under different attack scenarios. In direct, white-box attacks (Table A.4), the attacker tailors the prompts to the defended model, providing a more stringent test of our defense. In transfer attacks (Figure A.7), the prompts are not specifically optimized against our defense, resulting in different success rates.
> >
> > 2. **Definition and Evaluation of Adversarial Attacks**
> >
> >    *Response*: In our experiments, the attacks we evaluated—PAIR and AutoDAN—were conducted with the defense parameters fixed but without the attackers explicitly optimizing against our defense parameters. They are dynamic attacks that continuously modify their inputs based on the model's outputs but do not have explicit knowledge of our defense adjustments.
> >
> >    To address this, we have revised the manuscript to more precisely describe the nature of the attacks we evaluated. We have updated Appendix A.2.7 to refer to these attacks as **adversarial attacks** rather than **adaptive attacks**, acknowledging that while they are sophisticated and challenging, they may not fully represent an attacker explicitly optimizing against our defense parameters.
> >
> > 3. **Limitations and Discussion**
> >
> >    *Response*: We appreciate your insightful suggestion regarding fully adaptive attacks. Evaluating against such scenarios where the attacker explicitly optimizes against our defense parameters would indeed provide the most stringent test of robustness.
> >
> >    In fact, the scenario you described—where the attacker has complete knowledge of all safety directions ($\mathbf{d}_{safe}^l$), masks ($\mathbf{m}^l$), and the scaling factor ($\alpha$)—and can actively manipulate the inference process to set $\alpha<0$, is already partially addressed in our original results. Specifically, as highlighted in Section 5.3 (lines 518–521, marked for emphasis in the revised manuscript), when $\alpha<0$, the internal states are shifted in the opposite direction, effectively decreasing the safety of the model. This situation is illustrated in Figure 5 and Figure A.8, where the DSR rapidly decreases below 50% as $\alpha$ becomes negative.
> >
> >    This behavior highlights that if an attacker could fully manipulate the model's internal mechanisms, they could potentially undermine the defense. We acknowledge that such fully adaptive attacks represent a significant challenge for any defense, especially when the attacker has control over both the defense parameters and the inference process.
> >
> >    To address this, we have expanded the Conclusion in Section 5.4 and Appendix A.3 (Limitations and Future Work) to explicitly discuss this limitation. We hope this explanation clarifies our acknowledgment of this challenge and our perspective that fully adaptive attacks pose a fundamental difficulty for all defense strategies. Addressing these extreme adversarial scenarios remains a critical area for future research.
> >
> > ---
> >
> > We hope our responses have addressed your concerns and clarified key points. Your feedback has greatly improved our work, and we kindly ask you to reconsider your evaluation.

---

> > > ### Comment · Reviewer_FZR8 · 2024-11-26
> > >
> > > Thank you for the rebuttal efforts! I'm raising my score to 6.
> > >
> > > P.S.: I do believe that GCG (in your first point) can be counted as an adaptive attack, but PAIR and AutoDAN are not.

---

> > > > ### Author Response · Authors · 2024-11-27
> > > >
> > > > Thank you very much for your thoughtful feedback and for raising your score! Your comments on distinguishing adaptive and adversarial attacks have been incredibly helpful in refining our terminology and analysis. We're glad that our clarifications on GCG align with your perspective.
> > > >
> > > > If there’s anything else you believe we could address further, we’d be more than happy to do so. Thanks again for your engagement and valuable insights throughout this process!

---

### Official Review · Reviewer_gjnV · 2024-10-27

**Soundness:** 3
**Presentation:** 4
**Contribution:** 3
**Rating:** 8
**Confidence:** 3

**Summary:**

The paper propose a new defense against jailbreak attacks, called Jailbreak Antidote. It adjusts internal hidden state of LLMs during inference, with parameters controlling the tradeoff between safety and utility. Extensive experiments prove the performance of the method.

**Strengths:**

* The paper is looking at a well-defined hot security topic, the jailbreak problem.
* The paper is well written and easy to follow.
* The paper has a comprehensive evaluation, covering various state-of-the-art models, attack methods, mitigation baselines, etc., as well as ablation studies.

**Weaknesses:**

* The evaluation of false positive rate of safety blocking is missing in evaluation. See questions.
* I did not find implementation details of the PCA step.

**Questions:**

The paper is in a good shape and I enjoyed reading it. I have several questions on evaluation results.

The evaluation uses defense success rate against jailbreak attacks and win rate on benign prompts to demonstrate the balance between safety and utility. However, the defense success rate only represent the true positive rate of safety blocking, while false positive rate is missing. In other words, the statement in Section 4.3 "A higher \alpha emphasizes safety, making the model more conservative in its responses but potentially affecting utility by **increasing the refusal of borderline benign prompts**." is not validated by experiments. I would admit that the decent win rate on AlpacaEval has demonstrated that such borderline benign prompts is not common, but specific analysis on such failing cases is also meaningful. For instance, how much of the drop on win rate is because of false safety blocking?

Though the modification of internal state at inference stage should be very lightweight, I am unsure about the difficulty of the PCA step generating $d_{safe}$. How much data is used for PCA? How fast is the step especially for 70B large models?

---

> ### Author Response · Authors · 2024-11-22
> **Rebuttal - Part I/I**
>
> *We thank the reviewer for the positive feedback on our work and the acknowledgment of its relevance and comprehensive evaluation. We address your questions below.*
>
> 1. **Evaluation of False Positive Rate in Safety Blocking**
>
>     *Comment*: "The evaluation of false positive rate of safety blocking is missing in evaluation... how much of the drop on win rate is because of false safety blocking?"
>
>     *Response*: This is an important point. We have conducted an analysis of the false positive rate by examining the rate of inappropriate refusals in the model's responses on AlpacaEval. We define a response as a refusal if it begins with phrases like "I cannot," indicating an unnecessary rejection of a benign prompt [8]. We provide the refusal rates and corresponding win rates (Llama-3-70B-it & Llama-3-8B-it) for varying values of the scaling factor $\alpha$ in Figure A.9 of the appendix.
>
>     For example, for Llama-3-70B-it:
>
>     | $\alpha$        | 0.079    | 0.132    | 0.184    | 0.237    | 0.289    | 0.342    | 0.395    | 0.447    | 0.500    |
>     |--------------|----------|----------|----------|----------|----------|----------|----------|----------|----------|
>     | Refuse Rate  | 0.007    | 0.016    | 0.039    | 0.092    | 0.236    | 0.523    | 0.785    | 0.945    | 0.990    |
>     | Win Rate     | 55.050   | 55.839   | 55.963   | 53.487   | 41.677   | 22.919   | 7.205    | 0.745    | 0.124    |
>
>
>     This analysis confirms that higher values of $\alpha$ increase the model's conservativeness, leading to more refusals of benign prompts, and thus affecting utility. We have added this analysis to the revised manuscript (Appendix A.2.8).
>
>
> 2. **Difficulty of the PCA Step and Data Used**
>
>    *Comment*: "I am unsure about the difficulty of the PCA step generating d_safe. How much data is used for PCA? How fast is the step especially for 70B large models?"
>
>    *Response*: The PCA step is computationally lightweight. We used a small dataset of 128 pairs of benign and harmful prompts for PCA, ensuring minimal computational overhead. The pre-inference time for generating the safety directions is provided in Table A.6 of the appendix (`Table R.3` in the General Response). For instance, it takes approximately 5 minutes for a 70B model, which is negligible compared to fine-tuning methods. We have added implementation details in Eqs (4) & (5) in Section 4.1 of the revised manuscript.
>
> ---
>
> Thank you for highlighting important points about false positives and computational costs. Your feedback has refined our analysis and presentation. We trust the revised manuscript addresses your concerns and invite you to reconsider its evaluation. Thank you again for your valuable feedback.

---

> > ### Comment · Reviewer_gjnV · 2024-11-23
> > **Response to rebuttal**
> >
> > I read the revised version and my questions are addressed.

---

> > > ### Author Response · Authors · 2024-11-26
> > >
> > > Thank you for reviewing our revisions and confirming that your concerns have been addressed. We greatly appreciate your feedback and support!

---

### Official Review · Reviewer_EqZp · 2024-10-30

**Soundness:** 2
**Presentation:** 3
**Contribution:** 3
**Rating:** 6
**Confidence:** 4

**Summary:**

This paper studies an efficient defense, named Jailbreak Antidote, against jailbreak attacks targeting LLMs. The goals of defense design include balancing safety and utility, as well as reducing overhead during inference time. The paper is based on an observation drawn from the representations of harmful and safe prompts. Given the distance between these two representations, the authors modify the hidden state by add the masked distance vector to it to improve safety performance. Evaluations over nine LLMs are presented to demonstrate the results.

**Strengths:**

The tradeoff between LLM safety and utility is important. The paper is well-written and easy to follow. The authors showed some promising experimental results.

**Weaknesses:**

The paper misses comparison or discussion to related work. Additional experimental evaluations on more LLMs and specific downstream tasks should be performed to better support the claims in this paper. See Questions for more detail.

**Questions:**

I have some major comments on scope of related work and experiment design.
- The design of Jailbreak Antidote reads very similar to SafeDecoding [a], with one modifying the internal state while the other modifying the token distribution. However, I could not find any comparison or discussion with it. It would be great to see the experimental comparisons between these two defenses.
- Are the observations in A.3 consistent across all models? It is not clear which model was used to generate A.3.
- I wonder how would Jailbreak Antidote perform on uncensored models such as Dolphin-llama.
- The authors should include more attack methods such as [b] to evaluate the effectiveness of Jailbreak Antidote.
- Since utility is an important design goal, the authors should evaluate the model performance on more downstream tasks in addition to Win Rate, such as using WildBench, MMLU, and HellaSwag.
- Jailbreak Antidote seems to be sensitive to the choice of hyper-parameters. Is there any systematic way to choose these parameters, especially $\alpha$, for different models?

Minor comments:
- Although $d_{safe}^l$ and $m^l$ can be computed offline, it would be good to see the associated computational cost.
- Defense Tokens seem undefined.
- Is length controlled win rate also evaluated?
- In the notation of $h_S^l$, the subscript is not clear. The paper sometimes uses prompt as subscript, sometimes uses T as subscript.

[a] SafeDecoding: Defending against Jailbreak Attacks via Safety-Aware Decoding
[b] DeepInception: Hypnotize Large Language Model to Be Jailbreaker

---

> ### Author Response · Authors · 2024-11-22
> **Rebuttal - Part I/II**
>
> *We thank the reviewer for recognizing the importance of the safety-utility trade-off and the promising results of our method. We address your concerns as follows.*
>
>
> 1. **Comparison with Safe Decoding and Related Work**
>
>    *Comment*: "The design of Jailbreak Antidote reads very similar to SafeDecoding [a]... It would be great to see the experimental comparisons between these two defenses."
>
>    *Response*: Thank you for bringing this to our attention. While our method shares the goal of enhancing safety, it differs significantly from SafeDecoding [3]. Our method adjusts internal representations without requiring a reference model or additional decoding steps, leading to lower computational overhead. Safe Decoding relies on a fine-tuned reference model and modifies the decoding process, which can increase inference latency.
>
>    We have included experimental comparisons with Safe Decoding in the following table:
>
>    | **Defense / Model** (DSR / WinRate) | **Llama-2-7B-chat** | **Dolphin-Llama-2-7B** | **Llama-3-8B-it** |
>    |------------------|-|----|---------------------|
>    | Baseline                             | 35.6% / 50.0%       | 43.5% / 50.0%          | 68.9% / 50.0%      |
>    | Safe Decoding                        | 89.6% / 48.3%       | 78.6% / 46.8%          | 95.6% / 48.1%      |
>    | **Jailbreak Antidote**               | **92.3% / 51.6%**   | **87.3% / 52.0%**      | **99.4% / 53.3%**  |
>
>    Our method achieves higher defense success rates and better maintains utility across various models. Additionally, our method does not require fine-tuning a reference model, reducing the pre-inference cost, and introduces minimal inference overhead.
>
> 2. **Consistency of Observations Across Models**
>
>    *Comment*: "Are the observations in A.3 consistent across all models? It is not clear which model was used to generate A.3."
>
>    *Response*: We apologize for the oversight. Figure A.3 was generated using the Llama-3-8B-it model. We have updated the captions in the appendix to specify the models used. We have also included similar analyses for other models (Figures A.4, A.5, and A.6), showing that the observations are consistent across different architectures.
>
> 3. **Performance on Uncensored Models**
>
>    *Comment*: "I wonder how would Jailbreak Antidote perform on uncensored models such as Dolphin-llama."
>
>    *Response*: We have conducted experiments on the uncensored model Dolphin-Llama-2-7B. The results indicate that our method significantly enhances the safety of uncensored models, achieving high defense success rates while maintaining utility:
>
>    | **Method** | **Defense DSR on Dolphin-Llama-2-7B (%)** |
>    |-|---|
>    | Baseline            | 43.5    |
>    | In-Context Learning | 50.0   |
>    | Paraphrase          | 55.3  |
>    | PerplexityFilter    | 49.1   |
>    | SelfReminder        | 55.1  |
>    | Semantic Smooth     | 71.8    |
>    | SmoothLLM           | 48.5   |
>    | **Jailbreak Antidote** | **87.3**  |
>
>    Our method effectively enhances the safety of uncensored models.
>
>
> 4. **Including More Attack Methods**
>
>    *Comment*: "The authors should include more attack methods such as [b] to evaluate the effectiveness of Jailbreak Antidote."
>
>    *Response*: We have incorporated additional attack methods, including DeepInception [5], in our evaluation. The results demonstrate that our method remains effective against a wider range of attacks.
>
>    | **DeepInception** | **Gemma-2-2B-it** | **Phi-3-mini-it** | **Qwen-1.5-7B-it** | **Qwen-2-7B-it** | **Llama-3-8B-it** | **Llama-3.1-8B-it** | **Gemma-2-9B-it** |
>    |--|--|--|---|-|--|----|--|
>    | Baseline          | 46%               | 46%               | 0%                 | 56%              | 59%               | 62%                 | 46%               |
>    | In-Context Learning | 60%             | 37%               | 23%                | 54%              | 46%               | 46%                 | 68%               |
>    | Paraphrase        | 76%               | 76%               | 81%                | 79%              | 82%               | 89%                 | 91%               |
>    | PerplexityFilter  | 52%               | 51%               | 0%                 | 65%              | 53%               | 63%                 | 51%               |
>    | SelfReminder      | 63%               | 42%               | 16%                | 91%              | 68%               | 68%                 | 63%               |
>    | Semantic Smooth   | 78%               | 68%               | 76%                | 86%              | 92%               | 72%                 | 87%               |
>    | SmoothLLM         | 56%               | 82%               | 36%                | 72%              | 82%               | 71%                 | 27%               |
>    | **Jailbreak Antidote** | **81%**       | **93%**           | **84%**            | **98%**          | **100%**          | **93%**             | **92%**           |

---

> > ### Author Response · Authors · 2024-11-22
> > **Rebuttal - Part II/II**
> >
> > 5. **Evaluation on More Downstream Tasks**
> >
> >    *Comment*: "Since utility is an important design goal, the authors should evaluate the model performance on more downstream tasks in addition to Win Rate, such as using WildBench, MMLU, and HellaSwag."
> >
> >    *Response*: We agree that assessing utility on additional tasks is important. We have included evaluations on MMLU and HellaSwag datasets. The results, shown in the following table and Table A.8 of the appendix, confirm that our method maintains competitive performance on these tasks, supporting the claim that our method preserves utility.
> >
> >    | **Metric**   | **Baseline** | **In-Context Learning** | **Paraphrase** | **PerplexityFilter** | **SelfReminder** | **Semantic Smooth** | **SmoothLLM** | **Jailbreak Antidote** |
> >    |-|-------|----|---|-----|-|----|-------|-------|
> >    |AlpacaEvalWinRate(%)|50.0|38.9|35.5|52.2|39.4|31.8|32.4|**53.0**|
> >    |MMLUAccuracy(%)|66.7|65.3|64.3|66.7|65.8|62.8|63.2|**67.4**|
> >    |HellaSwagAccuracy(%)|82.2|80.1|79.5|82.3|78.6|80.6|73.9|**82.5**|
> >
> >    Our method maintains high performance on these downstream tasks, indicating that it preserves the model's utility.
> >
> >
> > 6. **Sensitivity to Hyperparameters**
> >
> >    *Comment*: "Jailbreak Antidote seems to be sensitive to the choice of hyper-parameters. Is there any systematic way to choose these parameters, especially $\alpha$, for different models?"
> >
> >    *Response*: We acknowledge that the choice of hyperparameters influences the balance between safety and utility. We consider this flexibility a feature, allowing users to adjust the model's safety preference at runtime based on their needs. The optimal $\alpha$ values typically fall within a manageable range (e.g., [0, 2]). We have retained the guidance in the manuscript on selecting $\alpha$ based on the desired trade-off, and included additional experiments illustrating the impact of $\alpha$ on different models (Figure A.8 in the appendix).
> >
> >
> > 7. **Computational Cost of Pre-Inference Steps**
> >
> >    *Comment*: "Although $d^l_{safe}$ and $m^l$ can be computed offline, it would be good to see the associated computational cost."
> >
> >    *Response*: We have included a detailed analysis of the pre-inference computational costs in Table A.6 of the appendix (`Table R.3` in the General Response). The results show that the pre-inference time is minimal, even for large models (e.g., approximately 5 minutes for a 70B model), making our method practical for deployment, especially compared to methods that require fine-tuning.
> >
> > 8. **Definition of Defense Tokens**
> >
> >    *Comment*: "Defense Tokens seem undefined."
> >
> >    *Response*: Thank you for pointing this out. We have added a clear definition of defense tokens in the revised manuscript (Section 5.2). Defense tokens refer to internal tokens used during the defense process, excluding the final tokens shown to the user. The number of these tokens significantly impacts Time to First Token (TTFT) and resource usage. This metric, independent of hardware platforms and inference engines, provides a consistent and convenient basis for comparison.
> >
> >
> > 9. **Length-Controlled Win Rate Evaluation**
> >
> >    *Comment*: "Is length controlled win rate also evaluated?"
> >
> >    *Response*: Yes, we have evaluated the length-controlled win rate and included the results in Table A.5 of the appendix. The findings indicate that the difference in win rate between length-controlled and non-length-controlled settings remains relatively small. Nonetheless, our proposed method, Jailbreak Antidote, consistently achieves a higher win rate under such controlled settings. This improvement may stem from the conservative nature of our defense strategy, which produces fewer but more consistent responses (favoring refusals to answer in the JailbreakBench benchmark), ensuring both security and effectiveness under length-controlled conditions.
> >
> >
> > 10. **Clarification on Notation**
> >
> >     *Comment*: "In the notation of $h_S^l$, the subscript is not clear. The paper sometimes uses prompt as subscript, sometimes uses T as subscript."
> >
> >     *Response*: We apologize for the confusion. In our notation, $h_S^l$ represents the hidden state at layer $l$ for the prompt $S$. We focus on the hidden state corresponding to the last token (position $T$), and in subsequent equations, we denote this as $h^l = h_T^l$. We clarified this in the manuscript (Line 173, Section 3).
> >
> > ---
> >
> > We appreciate your constructive feedback on comparisons, adaptive attacks, and additional benchmarks. Your suggestions prompted meaningful improvements in clarity and robustness. We hope the revisions meet your expectations and lead to a more favorable assessment.

---

> > > ### Comment · Reviewer_EqZp · 2024-11-23
> > >
> > > I appreciate the authors' efforts in addressing the comments. I will raise my score.

---

> > > > ### Author Response · Authors · 2024-11-26
> > > >
> > > > Thank you so much for your thoughtful feedback and for being open to updating your score. If there’s anything in the current version or something we can further clarify or improve that would help strengthen your evaluation, we’d be more than happy to address it right away.
> > > >
> > > > Your comments have already contributed significantly to refining our work, and we truly value the opportunity to make it even better. Thank you again for your time and consideration!

---

### Official Review · Reviewer_1gVh · 2024-11-03

**Soundness:** 3
**Presentation:** 3
**Contribution:** 2
**Rating:** 5
**Confidence:** 3

**Summary:**

This paper proposed a new defense to mitigate jailbreak attacks. The key idea of the proposed defense is to modify the internal state of an LLM during inference. The idea of the method is to extract a sparse safety feature vector and add it to the representation of a prompt. As a result, the model is more likely to refuse the malicious prompts.

**Strengths:**

1. How to defend against jailbreak attacks is still an open challenge in the community.

2. Multiple LLMs are evaluated.

3. The proposed attack does not rely on the attacks. Moreover, the method is efficient as it directly manipulates the internal states of an LLM during its inference.

**Weaknesses:**

1. In general, the compared baselines are very weak. The authors may consider doing a comprehensive survey on the defenses against jailbreak attacks and add a discussion on these defenses. For instance, the authors could check this survey papers: https://arxiv.org/pdf/2407.04295

2. The proposed method requires harmful prompts to identify safety direction. In the Appendix, it is mentioned that the dataset in Phan 2023 is used. Could authors evaluate the difference between this dataset and the dataset used for evaluation? For instance, whether there are many similar questions in the two datasets.

Additionally, a straightforward baseline is to use preference learning to finetune an LLM on this dataset. The authors may also consider comparing with this baseline in some settings (e.g., for small LLMs).

3. Adaptive attacks are not considered. If an attacker knows the safety vector (or a surrogate one optimized by the attacker), could an attacker perform adaptive attacks to evade the defense?

4. Only one benchmark dataset and 100 prompts are used in the evaluation. The authors may consider evaluating the effectiveness of the defense on more benchmark datasets.

**Questions:**

See weaknesses.

---

> ### Author Response · Authors · 2024-11-22
> **Rebuttal - Part I/II**
>
> *We thank the reviewer for highlighting the importance of defending against jailbreak attacks and acknowledging the efficiency of our method. We address each of your concerns below.*
>
> 1. **Comparison with More Comprehensive Baselines**
>
>    *Comment*: "In general, the compared baselines are very weak. The authors may consider doing a comprehensive survey on the defenses against jailbreak attacks and add a discussion on these defenses."
>
>    *Response*: We appreciate this suggestion. In the revised manuscript, we have expanded the related work section to include a comprehensive survey of defenses against jailbreak attacks, referencing the survey paper you mentioned [1]. Additionally, we have included experimental comparisons with advanced defense methods, including safety alignment defenses such as preference-based fine-tuning methods [7, 9] (Table 2 of the revised manuscript), and Safe Decoding [3] (Response to Reviewer `EqZp`). The results demonstrate that our method outperforms these baselines in both safety (DSR) and utility (WinRate) metrics. Furthermore, our method is compatible with these techniques and can enhance their performance when combined.
>
>
> 2. **Dataset Similarity Evaluation**
>
>    *Comment*: "The proposed method requires harmful prompts to identify safety direction... Could authors evaluate the difference between this dataset and the dataset used for evaluation? For instance, whether there are many similar questions in the two datasets."
>
>    *Response*: Thank you for pointing this out. We conducted a thorough similarity analysis between the dataset from Phan et al. (2023), used for extracting the safety directions, and the evaluation dataset (JailbreakBench). We employed metrics such as TF-IDF cosine similarity, n-gram Jensen-Shannon distance, and BERT cosine similarity. The results, detailed in Table A.1 of the appendix, indicate low similarity between the two datasets:
>
>    | **Metric**                                | **Value**   |
>    |---|-------------|
>    | TF-IDF Cosine Similarity                  | 0.038       |
>    | 1-gram & 2-gram Jensen-Shannon Distance   | 0.547       |
>    | BERT Cosine Similarity                    | 0.768       |
>
>    For reference, the BERT cosine similarity between the benign and harmful subsets within JailbreakBench is 0.840, which is higher than between the two datasets. This analysis confirms that there is no significant overlap that could bias our results. We have added this analysis to the revised manuscript (Appendix A.1.1).
>
>
>
> 3. **Comparison with Preference Learning Baselines**
>
>    *Comment*: "A straightforward baseline is to use preference learning to finetune an LLM on this dataset. The authors may also consider comparing with this baseline in some settings (e.g., for small LLMs)."
>
>    *Response*: We agree that comparing with preference learning baselines is valuable. We have included experiments using Direct Preference Optimization (DPO) as a baseline. The results, shown in the following table, demonstrate that our method achieves higher defense success rates (DSR) and better maintains utility (AlpacaEval WinRate) compared to models fine-tuned with DPO, especially on smaller models.
>
>    | **Model**          | **Baseline** (DSR / WinRate) | **DPO** (DSR / WinRate)    | **Antidote (Ours)** (DSR / WinRate) |
>    |-|----------|--------|-----------|
>    | Gemma-2-2B-it      | 29.2% / 50.0%                | 53.6% / 38.6%              | **71.8% / 52.0%**                   |
>    | Phi-3-mini-it      | 53.2% / 50.0%                | 83.3% / 23.6%              | **79.6% / 52.2%**                   |
>    | Qwen-1.5-7B-it     | 29.2% / 50.0%                | 58.4% / 39.6%              | **71.8% / 52.0%**                   |
>    | Qwen-2-7B-it       | 55.3% / 50.0%                | 83.6% / 41.7%              | **95.5% / 51.6%**                   |
>    | Llama-3-8B-it      | 68.9% / 50.0%                | 86.3% / 42.6%              | **99.4% / 53.0%**                   |
>    | Llama-3.1-8B-it    | 63.1% / 50.0%                | 72.6% / 36.7%              | **78.0% / 51.9%**                   |
>    | Gemma-2-9B-it      | 54.5% / 50.0%                | 67.4% / 37.6%              | **78.1% / 47.4%**                   |
>
>    Our method surpasses simple preference learning approaches, likely because we adjust only the neurons most relevant to safety, minimizing impact on the model's utility. Additionally, our method allows for real-time adjustment of the safety level without retraining, offering greater flexibility.

---

> > ### Author Response · Authors · 2024-11-22
> > **Rebuttal - Part II/II**
> >
> > 4. **Evaluation Against Adaptive Attacks**
> >
> >    *Comment*: "Adaptive attacks are not considered. If an attacker knows the safety vector... could an attacker perform adaptive attacks to evade the defense?"
> >
> >    *Response*: We appreciate this important point. We have added experiments evaluating our method against adaptive attacks, including GCG [8], PAIR [4], and AutoDAN [6]. The results, presented in Table A.4 of the appendix (`Table R.3` in the General Response), show that our method significantly improves defense success rates against these adaptive attacks across various models, demonstrating robustness even when attackers are aware of the defense mechanism.
> >
> >
> > 5. **Evaluation on More Benchmark Datasets**
> >
> >    *Comment*: "Only one benchmark dataset and 100 prompts are used in the evaluation. The authors may consider evaluating the effectiveness of the defense on more benchmark datasets."
> >
> >    *Response*: We have expanded our evaluation to include the AdvBench dataset [10], in addition to JailbreakBench. JailbreakBench-Behaviors contains a diverse set of behaviors, covering a wide range of misuse, and includes original content covering OpenAI's usage policies. AdvBench provides an additional test set to validate our method. The results, provided below, confirm that our method maintains superior performance across different datasets, further validating its effectiveness.
> >
> >    | **Defense / Model** | **Gemma-2-2B-it** | **Qwen-2-7B-it** | **Llama-3-8B-it** |
> >    |-|-------------------|------------------|-------------------|
> >    | Baseline            | 35.6%             | 63.3%            | 78.4%             |
> >    | In-Context Learning | 63.2%             | 67.4%            | 79.6%             |
> >    | Paraphrase          | 73.6%             | 78.6%            | 86.4%             |
> >    | PerplexityFilter    | 42.6%             | 62.4%            | 81.7%             |
> >    | SelfReminder        | 46.5%             | 80.6%            | 82.6%             |
> >    | Semantic Smooth     | 82.1%             | 86.7%            | 92.5%             |
> >    | SmoothLLM           | 62.3%             | 76.3%            | 90.1%             |
> >    | **Jailbreak Antidote** | **82.3%**        | **96.7%**        | **95.9%**         |
> >
> >    Our method achieves the highest defense success rates on AdvBench as well, demonstrating its generalizability and robustness.
> >
> > ---
> >
> > Thank you for your insightful comments, which helped us expand baseline comparisons, analyze dataset similarity, and enhance experimental rigor. We believe these revisions address your concerns and greatly improve the work. We kindly request a reevaluation of the manuscript.

---

> > > ### Author Response · Authors · 2024-11-26
> > > **Follow-Up on Author-Reviewer Discussion**
> > >
> > > Dear Reviewer 1gVh,
> > >
> > > Thank you again for your thoughtful feedback on our manuscript. We’ve worked hard to address the points you raised, and we wanted to check in as the discussion period is nearing its end. Please let us know if there are any remaining concerns or questions we can clarify. We’d greatly appreciate any further input to ensure the manuscript meets your expectations.
> > >
> > > Looking forward to hearing your thoughts!

---

### Author Response · Authors · 2024-11-22
**Rebuttal - Part I/II**

We sincerely thank all the reviewers for their thoughtful feedback and constructive comments on our manuscript titled "Jailbreak Antidote: Mitigating Jailbreak Attacks on LLMs via Sparse Runtime Adjustment of Internal States." We are grateful for the recognition of the importance of our work and the valuable suggestions provided to enhance its quality and clarity. We have carefully considered all comments and have revised the manuscript accordingly to address the concerns raised.


Several common themes emerged from the reviews, which we have carefully addressed in our revisions:

1. **Comparison with Additional Baselines and Related Work**: Reviewers suggested including comparisons with additional defense methods, particularly safety alignment defenses and preference-based fine-tuning approaches. In response, we have incorporated experimental comparisons with methods such as preference-based fine-tuning [7, 9, 3], demonstrating that our approach achieves superior defense success rates while maintaining better utility.

2. **Evaluation Against Adaptive Attacks**: Reviewers requested an assessment of the robustness of our method against adaptive attacks. To address this, we conducted experiments against adaptive attacks, including GCG [8], PAIR [4], and AutoDAN [6]. The results show that our method remains robust even when attackers are aware of our defense mechanism.

3. **Clarification on Dataset Similarity**: Concerns were raised about the potential overlap between the dataset used to generate safety directions and the evaluation dataset. We performed a thorough similarity analysis and confirmed that no significant overlap exists, ensuring unbiased evaluation.

4. **Additional Experiments and Analysis**: Reviewers recommended expanding experimental evaluations, covering more downstream tasks, and analyzing hyperparameter effects. In response, we included additional tasks such as MMLU and HellaSwag to assess utility, provided detailed analyses on hyperparameter impacts, and examined trade-offs between safety and utility, as well as false positive rates and efficiency.

5. **Clarification and Detail Enhancements**: Suggestions were made to improve clarity and provide more implementation details. We enhanced key sections for better readability, added implementation details to facilitate reproducibility, and addressed all minor issues, including notation and definitions.



---

**References**

1. Yi, Sibo, et al. "Jailbreak attacks and defenses against large language models: A survey." *arXiv preprint arXiv:2407.04295 (2024)*.

2. Wei, Boyi, et al. "Assessing the Brittleness of Safety Alignment via Pruning and Low-Rank Modifications." *Forty-first International Conference on Machine Learning*.

3. Xu, Zhangchen, et al. "SafeDecoding: Defending against Jailbreak Attacks via Safety-Aware Decoding." *ICLR 2024 Workshop on Secure and Trustworthy Large Language Models*.

4. Chao, Patrick, et al. "Jailbreaking Black Box Large Language Models in Twenty Queries." *R0-FoMo: Robustness of Few-shot and Zero-shot Learning in Large Foundation Models*.

5. Li, Xuan, et al. "Deepinception: Hypnotize large language model to be jailbreaker." *arXiv preprint arXiv:2311.03191 (2023)*.

6. Liu, Xiaogeng, et al. "AutoDAN: Generating Stealthy Jailbreak Prompts on Aligned Large Language Models." *The Twelfth International Conference on Learning Representations*.

7. Qi, X., et al. (2024). "Safety Alignment Should Be Made More Than Just a Few Tokens Deep." *arXiv preprint arXiv:2405.00159*.

8. Zou, A., et al. (2023). "Universal and Transferable Adversarial Attacks on Aligned Language Models." *arXiv preprint arXiv:2406.05946*.

9. Zou, Andy, et al. "Improving Alignment and Robustness with Short Circuiting." *arXiv preprint arXiv:2406.04313 (2024)*.

10. Chen, Yangyi, et al. "Why Should Adversarial Perturbations be Imperceptible? Rethink the Research Paradigm in Adversarial NLP." *Proceedings of the 2022 Conference on Empirical Methods in Natural Language Processing. 2022*.

---

> ### Author Response · Authors · 2024-11-22
> **Rebuttal - Part II/II**
>
> Below, we provide specific experimental results and details addressing the reviewers' concerns.
>
> **Comparisons with Safety Alignment Defenses** `Table R.1`
>
> We have included comparisons with safety alignment defenses, specifically preference-based fine-tuning methods. The results are summarized in the following table (Table 2 in the revised manuscript):
>
> | **Defense / Model**       | **Llama-3-8B-it** (DSR / WinRate) | **Llama-3-8B-it-RR** (DSR / WinRate) | **Gemma-2-9B-it** (DSR / WinRate) | **Gemma-2-9B-it-DSA** (DSR / WinRate) |
> |-----|-|-----|---|------|
> | Baseline                  | 68.9% / 50.0%         | 77.0% / 51.6%             | 54.5% / 50.0%           | 64.2% / 48.6%             |
> | In-Context Learning       | 71.7% / 38.9%         | 77.2% / 36.4%             | 56.7% / 38.6%           | 65.5% / 36.4%             |
> | Paraphrase                | 79.0% / 35.5%         | 91.1% / 32.6%             | 75.8% / 31.2%           | 81.1% / 28.6%             |
> | Perplexity Filter         | 67.9% / 52.2%         | 77.3% / 50.6%             | 55.1% / 51.0%           | 63.9% / 48.9%             |
> | Self Reminder             | 78.9% / 39.4%         | 80.5% / 42.2%             | 63.1% / 42.5%           | 69.5% / 39.0%             |
> | Semantic Smooth           | 88.1% / 31.8%         | 92.2% / 35.6%             | 79.4% / 33.9%           | 83.9% / 34.7%             |
> | SmoothLLM                 | 84.2% / 32.4%         | 95.6% / 31.5%             | 46.5% / 32.4%           | 51.7% / 32.8%             |
> | **Jailbreak Antidote**    | **99.4% / 53.0%**     | **99.6% / 53.5%**         | **78.1% / 47.4%**       | **83.6% / 48.6%**         |
>
> Our method outperforms the safety alignment defenses, achieving higher Defense Success Rates (DSR) while maintaining better utility (WinRate).
>
> **Evaluation Against Adaptive Attacks** `Table R.2`
>
> We evaluated our method against adaptive attacks, including GCG, PAIR, and AutoDAN. The results are summarized in the following table (Table A.4 in the appendix):
>
> | **Model**              | **GCG** (Baseline / Antidote) | **PAIR** (Baseline / Antidote) | **AutoDAN** (Baseline / Antidote) |
> |--|---------|---------|-------------|
> | Llama-2-7B-chat        | 46% / 83%                     | 78% / 91%                     | 67% / 86%                         |
> | Llama-2-13B-chat       | 70% / 86%                     | 85% / 93%                     | 76% / 87%                         |
> | Llama-3-8B-it          | 73% / 93%                     | 89% / 95%                     | 78% / 91%                         |
>
> Our method significantly improves defense success rates against these adaptive attacks, demonstrating robustness even when attackers are aware of the defense mechanism.
>
> **Pre-Inference Time** `Table R.3`
>
> We provide the pre-inference time for computing the safety directions for various models in the following table (Table A.6 in the appendix):
>
> | **Model**           | **Pre-Inference Time** |
> |----|--|
> | Gemma-2-2B-it       | 39.6 seconds           |
> | Phi-3-mini-it       | 43.2 seconds           |
> | Qwen-2-7B-it        | 48.2 seconds           |
> | Llama-3-8B-it       | 54.3 seconds           |
> | Gemma-2-9B-it       | 1 minute 26.4 seconds  |
> | Llama-3-70B-it      | 5 minutes 13.3 seconds |
> | Qwen-2-72B-it       | 5 minutes 32.5 seconds |
>
> The pre-inference computation is efficient, taking only a few minutes even for the largest models.
>
> **Inference Time Comparison** `Table R.4`
>
> We provide the actual inference time relative to the baseline (no defense) for our method in the following table (Table A.7 in the appendix):
>
> | **Model**           | **JailbreakBench** (Relative Time) | **AlpacaEval** (Relative Time) |
> |----|--------------|----------|
> | Gemma-2-2B-it       | 0.96x                              | 0.98x                          |
> | Phi-3-mini-it       | 0.94x                              | 0.97x                          |
> | Qwen-2-7B-it        | 0.85x                              | 0.98x                          |
> | Llama-3-8B-it       | 0.82x                              | 1.02x                          |
> | Gemma-2-9B-it       | 0.98x                              | 1.04x                          |
> | Llama-3-70B-it      | 0.79x                              | 1.02x                          |
> | Qwen-2-72B-it       | 0.86x                              | 1.01x                          |
>
> Our method introduces minimal additional computational cost during inference, often even reducing inference time due to shorter responses resulting from improved safety.
>
> ---
>
> We sincerely thank all reviewers for their valuable feedback, which has significantly improved the quality and clarity of our work. The revisions address key concerns and strengthen our contributions. We hope the updated manuscript resolves outstanding issues and merits a more favorable evaluation.

---

### Meta-Review · Area_Chair_798H · 2024-12-19

**Metareview:**

The submission received the ratings of four reviewers, which recommended 5, 6, 8 and 6, averaging 6.25. Given the plenty of competitive submissions in ICLR, this stands at a score above the borderline. The reviewers' concerns focus on the insufficient baselines, unclear technical parts or experimental setup, and the lack of analysis. After the substantial rebuttal by authors, most of reviewers considered the concerns well addressed, and one reviewer still maintain no response. After carefully checking the reviewer comments and the authors' feedback, the AC agreed with the authors' effort to address the remaining concerns. Therefore, I tend to recommend acceptance towards the current submission, and hope all advice are well incorporated into the final submission.

**Additional Comments On Reviewer Discussion:**

Please take all mentioned advice into the final revision to improve the submission.

---

### Decision · Program_Chairs · 2025-01-22

Accept (Poster)